# Active Regression by Stratification

**Sivan Sabato**
Department of Computer Science
Ben Gurion University, Beer Sheva, Israel
sabatos@cs.bgu.ac.il

**Remi Munos**[*]
INRIA
Lille, France
remi.munos@inria.fr

## Abstract

We propose a new active learning algorithm for parametric linear regression with random design. We provide finite sample convergence guarantees for general distributions in the misspecified model. This is the first active learner for this setting that provably can improve over passive learning. Unlike other learning settings (such as classification), in regression the passive learning rate of $O(1/\epsilon)$ cannot in general be improved upon. Nonetheless, the so-called 'constant' in the rate of convergence, which is characterized by a distribution-dependent *risk*, can be improved in many cases. For a given distribution, achieving the optimal risk requires prior knowledge of the distribution. Following the stratification technique advocated in Monte-Carlo function integration, our active learner approaches the optimal risk using piecewise constant approximations.

## 1 Introduction

In linear regression, the goal is to predict the real-valued labels of data points in Euclidean space using a linear function. The quality of the predictor is measured by the expected squared error of its predictions. In the standard regression setting with random design, the input is a labeled sample drawn i.i.d. from the joint distribution of data points and labels, and the cost of data is measured by the size of the sample. This model, which we refer to here as *passive learning*, is useful when both data and labels are costly to obtain. However, in domains where raw data is very cheap to obtain, a more suitable model is that of *active learning* (see, e.g., Cohn et al., 1994). In this model we assume that random data points are essentially free to obtain, and the learner can choose, for any observed data point, whether to ask also for its label. The cost of data here is the total number of requested labels.

In this work we propose a new active learning algorithm for linear regression. We provide finite sample convergence guarantees for general distributions, under a possibly misspecified model. For parametric linear regression, the sample complexity of passive learning as a function of the excess error $\epsilon$ is of the order $O(1/\epsilon)$. This rate cannot in general be improved by active learning, unlike in the case of classification (Balcan et al., 2009). Nonetheless, the so-called 'constant' in this rate of convergence depends on the distribution, and this is where the potential improvement by active learning lies.

Finite sample convergence of parametric linear regression in the passive setting has been studied by several (see, e.g., Györfi et al., 2002; Hsu et al., 2012). The standard approach is Ordinary Least Squares (OLS), where the output predictor is simply the minimizer of the mean squared error on the sample. Recently, a new algorithm for linear regression has been proposed (Hsu and Sabato, 2014). This algorithm obtains an improved convergence guarantee under less restrictive assumptions. An appealing property of this guarantee is that it provides a direct and tight relationship between the point-wise error of the optimal predictor and the convergence rate of the predictor. We exploit this to

---

[*]Current Affiliation: Google DeepMind.

allow our active learner to adapt to the underlying distribution. Our approach employs a stratification technique, common in Monte-Carlo function integration (see, e.g., Glasserman, 2004). For any finite partition of the data domain, an optimal oracle risk can be defined, and the convergence rate of our active learner approaches the rate defined by this risk. By constructing an infinite sequence of partitions that become increasingly refined, one can approach the globally optimal oracle risk.

Active learning for parametric regression has been investigated in several works, some of them in the context of statistical experimental design. One of the earliest works is Cohn et al. (1996), which proposes an active learning algorithm for locally weighted regression, assuming a well-specified model and an unbiased learning function. Wiens (1998, 2000) calculates a minimax optimal design for regression given the marginal data distribution, assuming that the model is approximately well-specified. Kanamori (2002) and Kanamori and Shimodaira (2003) propose an active learning algorithm that first calculates a maximum likelihood estimator and then uses this estimator to come up with an optimal design. Asymptotic convergence rates are provided under asymptotic normality assumptions. Sugiyama (2006) assumes an approximately well-specified model and i.i.d. label noise, and selects a design from a finite set of possibilities. The approach is adapted to pool-based active learning by Sugiyama and Nakajima (2009). Burbidge et al. (2007) propose an adaptation of Query By Committee. Cai et al. (2013) propose guessing the potential of an example to change the current model. Ganti and Gray (2012) propose a consistent pool-based active learner for the squared loss. A different line of research, which we do not discuss here, focuses on active learning for non-parameteric regression, e.g. Efromovich (2007).

**Outline** In Section 2 the formal setting and preliminaries are introduced. In Section 3 the notion of an *oracle risk* for a given distribution is presented. The stratification technique is detailed in Section 4. The new active learner algorithm and its analysis are provided in Section 5, with the main result stated in Theorem 5.1. In Section 6 we show via a simple example that in some cases the active learner approaches the maximal possible improvement over passive learning.

## 2 Setting and Preliminaries

We assume a data space in $\mathbb{R}^d$ and labels in $\mathbb{R}$. For a distribution $P$ over $\mathbb{R}^d \times \mathbb{R}$, denote by $\mathrm{supp}_X(P)$ the support of the marginal of $P$ over $\mathbb{R}^d$. Denote the strictly positive reals by $\mathbb{R}_+^*$. We assume that labeled examples are distributed according to a distribution $D$. A random labeled example is $(\mathbf{X}, Y) \sim D$, where $\mathbf{X} \in \mathbb{R}^d$ is the example and $Y \in \mathbb{R}$ is the label. Throughout this work, whenever $\mathbb{P}[\cdot]$ or $\mathbb{E}[\cdot]$ appear without a subscript, they are taken with respect to $D$. $D_X$ is the marginal distribution of $\mathbf{X}$ in pairs draws from $D$. The conditional distribution of $Y$ when the example is $\mathbf{X} = \mathbf{x}$ is denoted $D_{Y|\mathbf{x}}$. The function $\mathbf{x} \mapsto D_{Y|\mathbf{x}}$ is denoted $D_{Y|X}$.

A predictor is a function from $\mathbb{R}^d$ to $\mathbb{R}$ that predicts a label for every possible example. Linear predictors are functions of the form $\mathbf{x} \mapsto \mathbf{x}^\top \mathbf{w}$ for some $\mathbf{w} \in \mathbb{R}^d$. The squared loss of $\mathbf{w} \in \mathbb{R}^d$ for an example $\mathbf{x} \in \mathbb{R}^d$ with a true label $y \in \mathbb{R}$ is $\ell((\mathbf{x}, y), \mathbf{w}) = (\mathbf{x}^\top \mathbf{w} - y)^2$. The expected squared loss of $\mathbf{w}$ with respect to $D$ is $L(\mathbf{w}, D) = \mathbb{E}_{(\mathbf{X},Y) \sim D}[(\mathbf{X}^\top \mathbf{w} - Y)^2]$. The goal of the learner is to find a $\mathbf{w}$ such that $L(\mathbf{w})$ is small. The optimal loss achievable by a linear predictor is $L_\star(D) = \min_{\mathbf{w} \in \mathbb{R}^d} L(\mathbf{w}, D)$. We denote by $\mathbf{w}_\star(D)$ a minimizer of $L(\mathbf{w}, D)$ such that $L_\star(D) = L(\mathbf{w}_\star(D), D)$. In all these notations the parameter $D$ is dropped when clear from context.

In the passive learning setting, the learner draws random i.i.d. pairs $(\mathbf{X}, Y) \sim D$. The sample complexity of the learner is the number of drawn pairs. In the active learning setting, the learner draws i.i.d. examples $\mathbf{X} \sim D_X$. For any drawn example, the learner may draw a label according to the distribution $D_{Y|\mathbf{X}}$. The label complexity of the learner is the number of drawn labels. In this setting it is easy to approximate various properties of $D_X$ to any accuracy, with zero label cost. Thus we assume for simplicity direct access to some properties of $D_X$, such as the covariance matrix of $D_X$, denoted $\Sigma_D = \mathbb{E}_{\mathbf{X} \sim D_X}[\mathbf{X}\mathbf{X}^\top]$, and expectations of some other functions of $\mathbf{X}$. We assume w.l.o.g. that $\Sigma_D$ is not singular. For a matrix $A \in \mathbb{R}^{d \times d}$, and $\mathbf{x} \in \mathbb{R}^d$, denote $\|\mathbf{x}\|_A = \sqrt{\mathbf{x}^\top A \mathbf{x}}$. Let $R_D^2 = \max_{\mathbf{x} \in \mathrm{supp}_X(D)} \|\mathbf{x}\|_{\Sigma_D^{-1}}^2$. This is the *condition number* of the marginal distribution $D_X$. We have

$$\mathbb{E}[\|\mathbf{X}\|_{\Sigma_D^{-1}}^2] = \mathbb{E}[\mathrm{tr}(\mathbf{X}^\top \Sigma_D^{-1} \mathbf{X})] = \mathrm{tr}(\Sigma_D^{-1} \mathbb{E}[\mathbf{X}\mathbf{X}^\top]) = d. \qquad (1)$$

Hsu and Sabato (2014) provide a passive learning algorithm for least squares linear regression with a minimax optimal sample complexity (up to logarithmic factors). The algorithm is based on splitting the labeled sample into several subsamples, performing OLS on each of the subsamples, and then choosing one of the resulting predictors via a generalized median procedure. We give here a useful version of the result.[1]

**Theorem 2.1** (Hsu and Sabato, 2014). *There are universal constants $C, c, c', c'' > 0$ such that the following holds. Let $D$ be a distribution over $\mathbb{R}^d \times \mathbb{R}$. There exists an efficient algorithm that accepts as input a confidence $\delta \in (0, 1)$ and a labeled sample of size $n$ drawn i.i.d. from $D$, and returns $\hat{\mathbf{w}} \in \mathbb{R}^d$, such that if $n \geq c R_D^2 \log(c'n) \log(c''/\delta)$, with probability $1 - \delta$,*

$$L(\hat{\mathbf{w}}, D) - L_\star(D) = \|\mathbf{w}_\star(D) - \hat{\mathbf{w}}\|_{\Sigma_D}^2 \leq \frac{C \log(1/\delta)}{n} \cdot \mathbb{E}_D[\|\mathbf{X}\|_{\Sigma_D^{-1}}^2 (Y - \mathbf{X}^\top \mathbf{w}_\star(D))^2]. \quad (2)$$

This result is particularly useful in the context of active learning, since it provides an explicit dependence on the point-wise errors of the labels, including in heteroscedastic settings, where this error is not uniform. As we see below, in such cases active learning can potentially gain over passive learning. We denote an execution of the algorithm on a labeled sample $S$ by $\hat{\mathbf{w}} \leftarrow \text{REG}(S, \delta)$. The algorithm is used a black box, thus any other algorithm with similar guarantees could be used instead. For instance, similar guarantees might hold for OLS for a more restricted class of distributions.

Throughout the analysis we omit for readability details of integer rounding, whenever the effects are negligible. We use the notation $O(\exp)$, where $\exp$ is a mathematical expression, as a short hand for $\bar{c} \cdot \exp + \bar{C}$ for some universal constants $\bar{c}, \bar{C} \geq 0$, whose values can vary between statements.

## 3 An Oracle Bound for Active Regression

The bound in Theorem 2.1 crucially depends on the input distribution $D$. In an active learning framework, *rejection sampling* (Von Neumann, 1951) can be used to simulate random draws of labeled examples according to a different distribution, without additional label costs. By selecting a suitable distribution, it might be possible to improve over Eq. (2). Rejection sampling for regression has been explored in Kanamori (2002); Kanamori and Shimodaira (2003); Sugiyama (2006) and others, mostly in an asymptotic regime. Here we use the explicit bound in Eq. (2) to obtain new finite sample guarantees that hold for general distributions.

Let $\phi : \mathbb{R}^d \to \mathbb{R}_+^*$ be a strictly positive weight function such that $\mathbb{E}[\phi(\mathbf{X})] = 1$. We define the distribution $P_\phi$ over $\mathbb{R}^d \times \mathbb{R}$ as follows: For $\mathbf{x} \in \mathbb{R}^d, y \in \mathbb{R}$, let $\Gamma_\phi(\mathbf{x}, y) = \{(\tilde{\mathbf{x}}, \tilde{y}) \in \mathbb{R}^d \times \mathbb{R} \mid \mathbf{x} = \frac{\tilde{\mathbf{x}}}{\sqrt{\phi(\tilde{\mathbf{x}})}}, y = \frac{\tilde{y}}{\sqrt{\phi(\tilde{\mathbf{x}})}}\}$, and define $P_\phi$ by

$$\forall (\mathbf{X}, Y) \in \mathbb{R}^d \times \mathbb{R}, \qquad P_\phi(\mathbf{X}, Y) = \int_{(\tilde{\mathbf{X}}, \tilde{Y}) \in \Gamma_\phi(\mathbf{X}, Y)} \phi(\tilde{\mathbf{X}}) dD(\tilde{\mathbf{X}}, \tilde{Y}).$$

A labeled i.i.d. sample drawn according to $P_\phi$ can be simulated using rejection sampling without additional label costs (see Alg. 2 in Appendix B). We denote drawing $m$ random labeled examples according to $P$ by $S \leftarrow \text{SAMPLE}(P, m)$. For the squared loss on $P_\phi$ we have

$$\begin{aligned}
L(\mathbf{w}, P_\phi) &= \int_{(\mathbf{X}, Y) \in \mathbb{R}^d} \ell((\mathbf{X}, Y), \mathbf{w}) \, dP_\phi(\mathbf{X}, Y) \\
&\overset{(*)}{=} \int_{(\mathbf{X}, Y) \in \mathbb{R}^d} \ell((\mathbf{X}, Y), \mathbf{w}) \int_{(\tilde{\mathbf{X}}, \tilde{Y}) \in \Gamma_\phi(\mathbf{X}, Y)} \phi(\tilde{\mathbf{X}}) \, dD(\tilde{\mathbf{X}}, \tilde{Y}) \\
&= \int_{(\tilde{\mathbf{X}}, \tilde{Y}) \in \mathbb{R}^d} \ell((\frac{\tilde{\mathbf{X}}}{\sqrt{\phi(\tilde{\mathbf{X}})}}, \frac{\tilde{Y}}{\sqrt{\phi(\tilde{\mathbf{X}})}}), \mathbf{w}) \, \phi(\tilde{\mathbf{X}}) \, dD(\tilde{\mathbf{X}}, \tilde{Y}) \\
&= \int_{(\mathbf{X}, Y) \in \mathbb{R}^d} \ell((\mathbf{X}, Y), \mathbf{w}) \, dD(\mathbf{X}, Y) = L(\mathbf{w}, D).
\end{aligned}$$

The equality $(*)$ can be rigorously derived from the definition of Lebesgue integration. It follows that also $L_\star(D) = L_\star(P_\phi)$ and that $\mathbf{w}_\star(D) = \mathbf{w}_\star(P_\phi)$. We thus denote these by $L_\star$ and $\mathbf{w}_\star$. In

a similar manner, we have $\Sigma_{P_\phi} = \int \mathbf{X}\mathbf{X}^\top \, dP_\phi(\mathbf{X}, Y) = \int \mathbf{X}\mathbf{X}^\top \, dD(\mathbf{X}, Y) = \Sigma_D$. From now on we denote this matrix simply $\Sigma$. We denote $\|\cdot\|_\Sigma$ by $\|\cdot\|$, and $\|\cdot\|_{\Sigma^{-1}}$ by $\|\cdot\|_*$. The condition number of $P_\phi$ is $R_{P_\phi}^2 = \max_{\mathbf{x}\in\text{supp}_X(D)} \frac{\|\mathbf{x}\|_*^2}{\phi(\mathbf{x})}$.

If the regression algorithm is applied to $n$ labeled examples drawn from the simulated $P_\phi$, then by Eq. (2) and the equalities above, with probability $1 - \delta$, if $n \geq cR_{P_\phi}^2 \log(c'n)\log(c''/\delta))$,

$$L(\hat{\mathbf{w}}) - L_\star \leq \frac{C \cdot \log(1/\delta)}{n} \cdot \mathbb{E}_{P_\phi}[\|\mathbf{X}\|_*^2 (\mathbf{X}^\top \mathbf{w}_\star - Y)^2]$$

$$= \frac{C \cdot \log(1/\delta)}{n} \cdot \mathbb{E}_D[\|\mathbf{X}\|_*^2 (\mathbf{X}^\top \mathbf{w}_\star - Y)^2/\phi(\mathbf{X})].$$

Denote $\psi^2(\mathbf{x}) := \|\mathbf{x}\|_*^2 \cdot \mathbb{E}_D[(\mathbf{X}^\top \mathbf{w}_\star - Y)^2 \mid \mathbf{X} = \mathbf{x}]$. Further denote $\rho(\phi) := \mathbb{E}_D[\psi^2(\mathbf{X})/\phi(\mathbf{X})]$, which we term the *risk* of $\phi$. Then, if $n \geq cR_{P_\phi}^2 \log(c'n)\log(c''/\delta)$, with probability $1 - \delta$,

$$L(\hat{\mathbf{w}}) - L_\star \leq \frac{C \cdot \rho(\phi) \log(1/\delta)}{n}. \tag{3}$$

A passive learner essentially uses the default $\phi$, which is constantly 1, for a risk of $\rho(1) = \mathbb{E}[\psi^2(\mathbf{X})]$. But the $\phi$ that minimizes the bound is the solution to the following minimization problem:

$$\text{Minimize}_\phi \qquad \mathbb{E}[\psi^2(\mathbf{X})/\phi(\mathbf{X})]$$
$$\text{subject to} \qquad \mathbb{E}[\phi(\mathbf{X})] = 1, \tag{4}$$
$$\phi(\mathbf{x}) \geq \frac{c\log(c'n)\log(c''/\delta)}{n}\|\mathbf{x}\|_*^2, \quad \forall \mathbf{x} \in \text{supp}_X(D).$$

The second constraint is due to the requirement $n \geq cR_{P_\phi}^2 \log(c'n)\log(c''/\delta)$. The following lemma bounds the risk of the optimal $\phi$. Its proof is provided in Appendix C.

**Lemma 3.1.** *Let $\phi^\star$ be the solution to the minimization problem in Eq. (4). Then for $n \geq O(d\log(d)\log(1/\delta))$, $\mathbb{E}^2[\psi(\mathbf{X})] \leq \rho(\phi^\star) \leq \mathbb{E}^2[\psi(\mathbf{X})](1 + O(d\log(n)\log(1/\delta)/n))$.*

The ratio between the risk of $\phi^\star$ and the risk of the default $\phi$ thus approaches $\mathbb{E}[\psi^2(\mathbf{X})]/\mathbb{E}^2[\psi(\mathbf{X})]$, and this is also the optimal factor of label complexity reduction. The ratio is 1 for highly symmetric distributions, where the support of $D_X$ is on a sphere and all the noise variances are identical. In these cases, active learning is not helpful, even asymptotically. However, in the general case, this ratio is unbounded, and so is the potential for improvement from using active learning. The crucial challenge is that without access to the conditional distribution $D_{Y|X}$, Eq. (4) cannot be solved directly. We consider the *oracle risk* $\rho^\star = \mathbb{E}^2[\psi(\mathbf{X})]$, which can be approached if an oracle divulges the optimal $\phi$ and $n \to \infty$. The goal of the active learner is to approach the oracle guarantee *without* prior knowledge of $D_{Y|X}$.

## 4 Approaching the Oracle Bound with Strata

To approximate the oracle guarantee, we borrow the stratification approach used in Monte-Carlo function integration (e.g., Glasserman, 2004). Partition $\text{supp}_X(D)$ into $K$ disjoint subsets $\mathcal{A} = \{A_1, \ldots, A_K\}$, and consider for $\phi$ only functions that are constant on each $A_i$ and such that $\mathbb{E}[\phi(\mathbf{X})] = 1$. Each of the functions in this class can be described by a vector $\mathbf{a} = (a_1, \ldots, a_K) \in (\mathbb{R}_+^*)^K$. The value of the function on $\mathbf{x} \in A_i$ is $\frac{a_i}{\sum_{j\in[K]} p_j a_j}$, where $p_j := \mathbb{P}[\mathbf{X} \in A_j]$. Let $\phi_{\mathbf{a}}$ denote a function defined by $\mathbf{a}$, leaving the dependence on the partition $\mathcal{A}$ implicit. To calculate the risk of $\phi_{\mathbf{a}}$, denote $\mu_i := \mathbb{E}[\|\mathbf{X}\|_*^2 (\mathbf{X}^\top \mathbf{w}_\star - Y)^2 \mid \mathbf{X} \in A_i]$. From the definition of $\rho(\phi)$,

$$\rho(\phi_{\mathbf{a}}) = \sum_{j\in[K]} p_j a_j \sum_{i\in[K]} \frac{p_i}{a_i} \mu_i. \tag{5}$$

It is easy to verify that $\mathbf{a}^\star$ such that $a_i^\star = \sqrt{\mu_i}$ minimizes $\rho(\phi_{\mathbf{a}})$, and

$$\rho_{\mathcal{A}}^\star := \inf_{\mathbf{a}\in\mathbb{R}_+^K} \rho(\phi_{\mathbf{a}}) = \rho(\phi_{\mathbf{a}^\star}) = \Big(\sum_{i\in[K]} p_i\sqrt{\mu_i}\Big)^2. \tag{6}$$

$\rho_{\mathcal{A}}^\star$ is the oracle risk for the fixed partition $\mathcal{A}$. In comparison, the standard passive learner has risk $\rho(\phi_1) = \sum_{i\in[K]} p_i \mu_i$. Thus, the ratio between the optimal risk and the default risk can be as large as $1/\min_i p_i$. Note that here, as in the definition of $\rho^\star$ above, $\rho_{\mathcal{A}}^\star$ might not be achievable for samples up to a certain size, because of the additional requirement that $\phi$ not be too small (see Eq. (4)). Nonetheless, this optimistic value is useful as a comparison.

Consider an infinite sequence of partitions: for $j \in \mathbb{N}$, $\mathcal{A}^j = \{A_1^j, \ldots, A_{K_j}^j\}$, with $K_j \to \infty$. Similarly to Carpentier and Munos (2012), under mild regularity assumptions, if the partitions have diameters and probabilities that approach zero, then $\rho_{\mathcal{A}^j}^\star \to \rho(\phi^\star)$, achieving the optimal upper bound for Eq. (3). For a fixed partition $\mathcal{A}$, the challenge is then to approach $\rho_{\mathcal{A}}^*$ without prior knowledge of the true $\mu_i$'s, using relatively few extra labeled examples. In the next section we describe our active learning algorithm that does just that.

## 5 Active Learning for Regression

To approach the optimal risk $\rho_{\mathcal{A}}^*$, we need a good estimate of $\mu_i$ for $i \in [K]$. Note that $\mu_i$ depends on the optimal predictor $\mathbf{w}_\star$, therefore its value depends on the entire distribution. We assume that the error of the label relative to the optimal predictor is bounded as follows: There exists a $b \geq 0$ such that $(\mathbf{x}^\top \mathbf{w}_\star - y)^2 \leq b^2 \|\mathbf{x}\|_*^2$ for all $(\mathbf{x}, y)$ in the support of $D$. This boundedness assumption can be replaced by an assumption on sub-Gaussian tails with similar results. Our assumption implies also $L_\star = \mathbb{E}[(\mathbf{x}^\top \mathbf{w}_\star - y)^2] \leq b^2 \mathbb{E}[\|\mathbf{X}\|_*^2] = b^2 d$, where the last equality follows from Eq. (1).

---

**Algorithm 1** Active Regression

**input** Confidence $\delta \in (0, 1)$, label budget $m$, partition $\mathcal{A}$.
**output** $\hat{\mathbf{w}} \in \mathbb{R}^d$
1:   $m_1 \leftarrow m^{4/5}/2$, $m_2 \leftarrow m^{4/5}/2$, $m_3 \leftarrow m - (m_1 + m_2)$.
2:   $\delta_1 \leftarrow \delta/4$, $\delta_2 \leftarrow \delta/4$, $\delta_3 \leftarrow \delta/2$.
3:   $S_1 \leftarrow \text{SAMPLE}(P_{\phi[\Sigma]}, m_1)$
4:   $\hat{\mathbf{v}} \leftarrow \text{REG}(S_1, \delta_1)$
5:   $\Delta \leftarrow \sqrt{\frac{Cd^2 b^2 \log(1/\delta_1)}{m_1}}$;   $\gamma \leftarrow (b + 2\Delta)^2 \sqrt{K \log(2K/\delta_2)/m_2}$;   $t \leftarrow m_2/K$.
6: **for** $i = 1$ to $K$ **do**
7:     $T_i \leftarrow \text{SAMPLE}(Q_i, t)$.
8:     $\tilde{\mu}_i \leftarrow \Theta_i \cdot \left( \frac{1}{t} \sum_{(\mathbf{x},y)\in T_i} (|\mathbf{x}^\top \hat{\mathbf{v}} - y| + \Delta)^2 + \gamma \right)$.
9:     $\hat{a}_i \leftarrow \sqrt{\tilde{\mu}_i}$.
10: **end for**
11: $\xi \leftarrow \frac{c \log(c' m_3) \log(c''/\delta_3)}{m_3}$
12: Set $\hat{\phi}$ such that for $\mathbf{x} \in A_i$, $\hat{\phi}(\mathbf{x}) := \|\mathbf{x}\|_*^2 \cdot \xi + (1 - d\xi)\frac{\hat{a}_i}{\sum_j p_j \hat{a}_j}$.
13: $S_3 \leftarrow \text{SAMPLE}(P_{\hat{\phi}}, m_3)$.
14: $\hat{\mathbf{w}} \leftarrow \text{REG}(S_3, \delta_3)$.

---

Our active regression algorithm, listed in Alg. 1, operates in three stages. In the first stage, the goal is to find a crude loss optimizer $\hat{\mathbf{v}}$, so as to later estimate $\mu_i$. To find this optimizer, the algorithm draws a labeled sample of size $m_1$ from the distribution $P_{\phi[\Sigma]}$, where $\phi[\Sigma](\mathbf{x}) := \frac{1}{d}\mathbf{x}^\top \Sigma^{-1}\mathbf{x} = \frac{1}{d}\|\mathbf{x}\|_*^2$. Note that $\rho(\phi[\Sigma]) = d \cdot \mathbb{E}[(\mathbf{X}\mathbf{w}_\star - Y)^2] = dL_\star$. In addition, $R_{P_{\phi[\Sigma]}}^2 = d$. Consequently, by Eq. (3), applying REG to $m_1 \geq O(d \log(d) \log(1/\delta_1))$ random draws from $P_{\phi[\Sigma]}$ gets, with probability $1 - \delta_1$

$$L(\hat{\mathbf{v}}) - L_\star = \|\hat{\mathbf{v}} - \mathbf{w}_\star\|^2 \leq \frac{CdL_\star \log(1/\delta_1)}{m_1} \leq \frac{Cd^2 b^2 \log(1/\delta_1)}{m_1}. \quad (7)$$

In Needell et al. (2013) a similar distribution is used to speed up gradient descent for convex losses. Here, we make use of $\phi[\Sigma]$ as a stepping stone in order to approach the optimal $\phi$ at a rate that does not depend on the condition number of $D$. Denote by $\mathcal{E}$ the event that Eq. (7) holds.

In the second stage, estimates for $\mu_i$, denoted $\tilde{\mu}_i$, are calculated from labeled samples that are drawn from another set of probability distributions, $Q_i$ for $i \in [K]$. These distributions are defined as follows. Denote $\Theta_i = \mathbb{E}[\|\mathbf{X}\|_*^4 \mid \mathbf{X} \in A_i]$. For $\mathbf{x} \in \mathbb{R}^d, y \in \mathbb{R}$, let $\Gamma_i(\mathbf{x}, y) = \{(\tilde{\mathbf{x}}, \tilde{y}) \in A_i \times$

$\mathbb{R} \mid \mathbf{x} = \frac{\tilde{\mathbf{x}}}{\|\tilde{\mathbf{x}}\|_*}, y = \frac{\tilde{y}}{\|\tilde{\mathbf{x}}\|_*}\}$, and define $Q_i$ by $dQ_i(\mathbf{X}, Y) = \frac{1}{\Theta_i} \int_{(\tilde{\mathbf{X}}, \tilde{Y}) \in \Gamma_i(\mathbf{X}, Y)} \|\tilde{\mathbf{X}}\|_*^4 \, dD(\tilde{\mathbf{X}}, \tilde{Y})$.
Clearly, for all $\mathbf{x} \in \mathrm{supp}_X(Q_i)$, $\|\mathbf{x}\|_* = 1$. Drawing labeled examples from $Q_i$ can be done using rejection sampling, similarly to $P_\phi$. The use of the $Q_i$ distributions in the second stage again helps avoid a dependence on the condition number of $D$ in the convergence rates.

In the last stage, a weight function $\hat{\phi}$ is determined based on the estimated $\tilde{\mu}_i$. A labeled sample is drawn from $P_{\hat{\phi}}$, and the algorithm returns the predictor resulting from running REG on this sample. The following theorem gives our main result, a finite sample convergence rate guarantee.

**Theorem 5.1.** *Let $b \geq 0$ such that $(\mathbf{x}^\top \mathbf{w}_\star - y)^2 \leq b^2 \|\mathbf{x}\|_*^2$ for all $(\mathbf{x}, y)$ in the support of $D$. Let $\Lambda_D = \mathbb{E}[\|\mathbf{X}\|_*^4]$. If Alg. 1 is executed with $\delta$ and $m$ such that $m \geq O(d \log(d) \log(1/\delta))^{5/4}$, then it draws $m$ labels, and with probability $1 - \delta$,*

$$L(\hat{\mathbf{w}}) - L_\star \leq \frac{C \rho_{\mathcal{A}}^\star \log(3/\delta)}{m} +$$

$$O \left( \frac{\log(1/\delta)}{m^{6/5}} \rho_{\mathcal{A}}^\star + \frac{d^{1/2} \Lambda_D^{1/4} \log^{5/4}(1/\delta)}{m^{6/5}} b^{1/2} \rho_{\mathcal{A}}^{\star\,3/4} + \frac{d \Lambda_D^{1/2} K^{1/4} \log^{1/4}(K/\delta) \log(1/\delta)}{m^{6/5}} b \rho_{\mathcal{A}}^{\star\,1/2} \right).$$

The theorem shows that the learning rate of the active learner approaches the oracle rate for the given partition. With an infinite sequence of partitions with $K$ an increasing function of $m$, the optimal oracle risk can also be approached. The rate of convergence to the oracle rate does not depend on the condition number of $D$, unlike the passive learning rate. In addition, $m = O(d \log(d) \log(1/\delta))^{5/4}$ suffices to approach the optimal rate, whereas $m = \Omega(d)$ is obviously necessary for any learner. It is interesting that also in active learning for classification, it has been observed that active learning in a non-realizable setting requires a super-linear dependence on $d$ (See, e.g., Dasgupta et al., 2008). Whether this dependence is unavoidable for active regression is an open question. Theorem 5.1 is be proved via a series of lemmas. First, we show that if $\tilde{\mu}_i$ is a good approximation of $\mu_i$ then $\rho_{\mathcal{A}}(\hat{\phi})$ can be bounded as a function of the oracle risk for $\mathcal{A}$.

**Lemma 5.2.** *Suppose $m_3 \geq O(d \log(d) \log(1/\delta_3))$, and let $\hat{\phi}$ as in Alg. 1. If, for some $\alpha, \beta \geq 0$,*

$$\mu_i \leq \tilde{\mu}_i \leq \mu_i + \alpha_i \sqrt{\mu_i} + \beta_i, \tag{8}$$

*then*

$$\rho_{\mathcal{A}}(\hat{\phi}) \leq (1 + O(d \log(m_3) \log(1/\delta_3)/m_3))(\rho_{\mathcal{A}}^\star + (\sum_i p_i \alpha_i)^{1/2} \rho_{\mathcal{A}}^{\star\,3/4} + (\sum_i p_i \beta_i)^{1/2} \rho_{\mathcal{A}}^{\star\,1/2}).$$

*Proof.* We have $\forall \mathbf{x} \in A_i, \hat{\phi}(\mathbf{x}) \geq (1 - d\xi) \frac{\hat{a}_i}{\sum_j p_j \hat{a}_j}$, where $\xi = \frac{c \log(c' m_3) \log(c''/\delta)}{m_3}$. Therefore

$$\rho(\hat{\phi}) \equiv \mathbb{E}[\psi^2(\mathbf{X})/\hat{\phi}(\mathbf{X})] \leq \frac{1}{1 - d\xi} \sum_j p_j \hat{a}_j \sum_i p_i \cdot \mathbb{E}[\psi^2(\mathbf{X})/\hat{a}_i \mid \mathbf{X} \in A_i]$$

$$= \frac{1}{1 - d\xi} \sum_j p_j \hat{a}_j \sum_i p_i \mu_i / \hat{a}_i = (1 + \frac{d\xi}{1 - d\xi}) \rho(\phi_{\hat{\mathbf{a}}}).$$

For $m_3 \geq O(d \log(d) \log(1/\delta_3))$, $d\xi \leq \frac{1}{2}$,[2] therefore $\frac{d\xi}{1 - d\xi} \leq 2 d\xi$. It follows

$$\rho(\hat{\phi}) \leq (1 + O(d \log(m_3) \log(1/\delta_3)/m_3)) \rho(\phi_{\hat{\mathbf{a}}}). \tag{9}$$

By Eq. (8),

$$\rho_{\mathcal{A}}(\phi_{\hat{\mathbf{a}}}) = \sum_j p_j \sqrt{\tilde{\mu}_j} \sum_i p_i \mu_i / \sqrt{\tilde{\mu}_i}$$

$$\leq \sum_j p_j (\sqrt{\mu_j} + \sqrt{\alpha_j} \mu_j^{1/4} + \sqrt{\beta_j}) \sum_i p_i \sqrt{\mu_i}$$

$$= (\sum_i p_i \sqrt{\mu_i})^2 + (\sum_j p_j \sqrt{\alpha_j} \mu_j^{1/4})(\sum_i p_i \sqrt{\mu_i}) + (\sum_j p_j \sqrt{\beta_j})(\sum_i p_i \sqrt{\mu_i}).$$

$$= \rho_{\mathcal{A}}^\star + (\sum_j p_j \sqrt{\alpha_j} \mu_j^{1/4}) \rho_{\mathcal{A}}^{\star\,1/2} + (\sum_j p_j \sqrt{\beta_j}) \rho_{\mathcal{A}}^{\star\,1/2}.$$

The last equality is since $\rho_{\mathcal{A}}^\star = (\sum_i p_i \sqrt{\mu_i})^2$. By Cauchy-Schwartz, $(\sum_j p_j \sqrt{\alpha_j} \mu_j^{1/4}) \leq (\sum_i p_i \alpha_i)^{1/2} \rho_{\mathcal{A}}^{\star\,3/4}$. By Jensen's inequality, $\sum_j p_j \sqrt{\beta_j} \leq (\sum_j p_j \beta_j)^{1/2}$. Combined with Eq. (6) and Eq. (9), the lemma directly follows. $\qquad\square$

We now show that Eq. (8) holds and provide explicit values for $\alpha$ and $\beta$. Define

$$\nu_i := \Theta_i \cdot \mathbb{E}_{Q_i}[(|\mathbf{X}^\top \hat{\mathbf{w}} - Y| + \Delta)^2], \quad \text{and} \quad \hat{\nu}_i := \frac{\Theta_i}{t} \sum_{(\mathbf{x},y) \in T_i} (|\mathbf{x}^\top \hat{\mathbf{w}} - y| + \Delta)^2.$$

Note that $\tilde{\mu}_i = \hat{\nu}_i + \Theta_i \gamma$. We will relate $\hat{\nu}_i$ to $\nu_i$, and then $\nu_i$ to $\mu_i$, to conclude a bound of the form in Eq. (8) for $\tilde{\mu}_i$. First, note that if $m_1 \geq O(d \log(d) \log(1/\delta_1))$ and $\mathcal{E}$ holds, then for any $\mathbf{x} \in \cup_{i \in [K]} \mathrm{supp}_X(Q_i)$,

$$|\mathbf{x}^\top \hat{\mathbf{v}} - \mathbf{x}^\top \mathbf{w}_\star| \leq \|\mathbf{x}\|_* \|\hat{\mathbf{v}} - \mathbf{w}_\star\| \leq \sqrt{\frac{Cd^2 b^2 \log(1/\delta_1)}{m_1}} \equiv \Delta. \tag{10}$$

The second inequality stems from $\|\mathbf{x}\|_* = 1$ for $\mathbf{x} \in \cup_{i \in [K]} \mathrm{supp}_X(Q_i)$, and Eq. (7). This is useful in the following lemma, which relates $\hat{\nu}_i$ with $\nu_i$.

**Lemma 5.3.** *Suppose that $m_1 \geq O(d \log(d) \log(1/\delta_1))$ and $\mathcal{E}$ holds. Then with probability $1 - \delta_2$ over the draw of $T_1, \ldots, T_K$, for all $i \in [K]$, $|\hat{\nu}_i - \nu_i| \leq \Theta_i (b + 2\Delta)^2 \sqrt{K \log(2K/\delta_2)/m_2} \equiv \Theta_i \gamma$.*

*Proof.* For a fixed $\hat{\mathbf{v}}$, $\hat{\nu}_i / \Theta_i$ is the empirical average of i.i.d. samples of the random variable $Z = (|\mathbf{X}^\top \hat{\mathbf{v}} - Y| + \Delta)^2$, where $(\mathbf{X}, Y)$ is drawn according to $Q_i$. We now give an upper bound for $Z$ with probability 1. Let $(\tilde{\mathbf{X}}, \tilde{Y})$ in the support of $D$ such that $\mathbf{X} = \tilde{\mathbf{X}}/\|\tilde{\mathbf{X}}\|_*$ and $Y = \tilde{Y}/\|\tilde{\mathbf{X}}\|_*$. Then $|\mathbf{X}^\top \mathbf{w}_\star - Y| = |\tilde{\mathbf{X}}^\top \mathbf{w}_\star - \tilde{Y}|/\|\tilde{\mathbf{X}}\|_* \leq b$. If $\mathcal{E}$ holds and $m_1 \geq O(d \log(d) \log(1/\delta_1))$,

$$Z \leq (|\mathbf{X}^\top \hat{\mathbf{v}} - \mathbf{X}^\top \mathbf{w}_\star| + |\mathbf{X}^\top \mathbf{w}_\star - Y| + \Delta)^2 \leq (b + 2\Delta)^2,$$

where the last inequality follows from Eq. (10). By Hoeffding's inequality, for every $i$, with probability $1 - \delta_2$, $|\hat{\nu}_i - \nu_i| \leq \Theta_i (b + 2\Delta)^2 \sqrt{\log(2/\delta_2)/t}$. The statement of the lemma follows from a union bound over $i \in [K]$ and $t = m_2/K$. $\qquad\square$

The following lemma, proved in Appendix D, provides the desired relationship between $\nu_i$ and $\mu_i$.

**Lemma 5.4.** *If $m_1 \geq O(d \log(d) \log(1/\delta_1))$ and $\mathcal{E}$ holds, then $\mu_i \leq \nu_i \leq \mu_i + 4\Delta\sqrt{\Theta_i \mu_i} + 4\Delta^2 \Theta_i$.*

We are now ready to prove Theorem 5.1.

*Proof of Theorem 5.1.* From the condition on $m$ and the definition of $m_1, m_3$ in Alg. 1 we have $m_1 \geq O(d \log(d/\delta_1))$ and $m_3 \geq O(d \log(d/\delta_3))$. Therefore the inequalities in Lemma 5.4, Lemma 5.3 and Eq. (3) (with $n, \delta, \phi$ substituted with $m_3, \delta_3, \hat{\phi}$) hold simultaneously with probability $1 - \delta_1 - \delta_2 - \delta_3$. For Eq. (3), note that $\frac{\|\mathbf{x}\|_*}{\hat{\phi}(\mathbf{x})} \geq \xi$, thus $m_3 \geq c R_{P_{\hat{\phi}}}^2 \log(c'n) \log(c''/\delta_3)$ as required.

Combining Lemma 5.4 and Lemma 5.3, and noting that $\tilde{\mu}_i = \hat{\nu}_i + \Theta_i \gamma$, we conclude that

$$\mu_i \leq \tilde{\mu}_i \leq \mu_i + 4\Delta\sqrt{\Theta_i \mu_i} + \Theta_i(4\Delta^2 + 2\gamma).$$

By Lemma 5.2, it follows that

$$\rho_{\mathcal{A}}(\hat{\phi}) \leq \rho_{\mathcal{A}}^\star + 2\sqrt{\Delta} \Big( \sum_{i \in [K]} p_i \sqrt{\Theta_i} \Big)^{1/2} \rho_{\mathcal{A}}^{\star\,3/4} + \sqrt{4\Delta^2 + 2\gamma} \cdot \Big( \sum_{i \in [K]} p_i \Theta_i \Big)^{1/2} \rho_{\mathcal{A}}^{\star\,1/2} + \bar{O}\Big(\frac{\log(m_3)}{m_3}\Big)$$

$$\leq \rho_{\mathcal{A}}^\star + 2\Delta^{1/2} \Lambda_D^{1/4} \rho_{\mathcal{A}}^{\star\,3/4} + \sqrt{4\Delta^2 + 2\gamma} \cdot \Lambda_D^{1/2} \rho_{\mathcal{A}}^{\star\,1/2} + \bar{O}(\log(m_3)/m_3).$$

The last inequality follows since $\sum_{i \in [K]} p_i \Theta_i = \Lambda_D$. We use $\bar{O}$ to absorb parameters that already appear in the other terms of the bound. Combining this with Eq. (3),

$$L(\hat{\mathbf{w}}) - L_\star \leq \frac{C \rho_{\mathcal{A}}^\star \log(1/\delta_3)}{m_3} +$$

$$\frac{C \log(1/\delta_3)}{m_3} \Big( 2\Delta^{1/2} \Lambda_D^{1/4} \rho_{\mathcal{A}}^{\star\,3/4} + (2\Delta + \sqrt{2\gamma}) \cdot \Lambda_D^{1/2} \rho_{\mathcal{A}}^{\star\,1/2} \Big) + \bar{O}\Big(\frac{\log(m_3)}{m_3^2}\Big).$$

We have $\gamma = (b+2\Delta)^2 \sqrt{K \log(2K/\delta_2)/m_2}$, and $\Delta = \sqrt{\frac{Cd^2b^2\log(1/\delta_1)}{m_1}}$. For $m_1 \geq Cd\log(1/\delta_1)$, $\Delta \leq b\sqrt{d}$, thus $\gamma \leq b^2(2\sqrt{d}+1)^2 \sqrt{K\log(2K/\delta_2)/m_2}$. Substituting for $\Delta$ and $\gamma$, we have

$$L(\hat{\mathbf{w}}) - L_\star \leq \frac{C\rho_\mathcal{A}^\star \log(1/\delta_3)}{m_3} + \frac{C\log(1/\delta_3)}{m_3}\left(\frac{16Cd^2b^2\log(1/\delta_1)}{m_1}\right)^{1/4} \Lambda_D^{1/4} {\rho_\mathcal{A}^\star}^{3/4}$$

$$+ \frac{C\log(1/\delta_3)}{m_3}\left(\left(\frac{4Cd^2b^2\log(1/\delta_1)}{m_1}\right)^{1/2}\right.$$

$$\left.+ \sqrt{2}b(2\sqrt{d}+1)\left(\frac{K\log(2K/\delta_2)}{m_2}\right)^{1/4}\right)\cdot \Lambda_D^{1/2}{\rho_\mathcal{A}^\star}^{1/2} + \bar{O}(\frac{\log(m_3)}{m_3^2}).$$

To get the theorem, set $m_3 = m - m^{4/5}$, $m_2 = m_1 = m^{4/5}/2$, $\delta_1 = \delta_2 = \delta/4$, and $\delta_3 = \delta/2$. $\quad\square$

## 6 Improvement over Passive Learning

Theorem 5.1 shows that our active learner approaches the oracle rate, which can be strictly faster than the rate implied by Theorem 2.1 for passive learning. To complete the picture, observe that this better rate cannot be achieved by *any* passive learner. This can be seen by the following 1-dimensional example. Let $\sigma > 0, \alpha > \frac{1}{\sqrt{2}}, p = \frac{1}{2\alpha^2}$, and $\eta \in \mathbb{R}$ such that $|\eta| \leq \frac{\sigma}{\alpha}$. Let $D_\eta$ over $\mathbb{R} \times \mathbb{R}$ such that with probability $p$, $X = \alpha$ and $Y = \alpha\eta + \epsilon$, where $\epsilon \sim N(0, \sigma^2)$, and with probability $1-p$, $X = \beta := \sqrt{\frac{1-p\alpha^2}{1-p}}$ and $Y = 0$. Then $\mathbb{E}[X^2] = 1$ and $w_\star = p\alpha^2\eta$. Consider a partition of $\mathbb{R}$ such that $\alpha \in A_1$ and $\beta \in A_2$. Then $p_1 = p$, $\mu_1 = \mathbb{E}_\epsilon[\alpha^2(\epsilon + \alpha\eta - \alpha w_\star)^2] = \alpha^2(\sigma^2 + \alpha^2\eta^2(1 - p\alpha^2)) \leq \frac{3}{2}\alpha^2\sigma^2$. In addition, $p_2 = 1 - p$ and $\mu_2 = \beta^4 w_\star^2 = (\frac{1-p\alpha^2}{1-p})^2 p^2\alpha^4\eta^2 \leq \frac{p^2\alpha^2\sigma^2}{4(1-p)^2}$. The oracle risk is

$$\rho_\mathcal{A}^\star = (p_1\sqrt{\mu_1} + p_2\sqrt{\mu_2})^2 \leq (p\sqrt{\frac{3}{2}}\alpha\sigma + (1-p)\frac{p\alpha\sigma}{2(1-p)})^2 = p^2\alpha^2\sigma^2(\sqrt{\frac{3}{2}} + \frac{1}{2})^2 \leq 2p\sigma^2.$$

Therefore, for the active learner, with probability $1 - \delta$,

$$L(\hat{w}) - L_\star \leq \frac{2Cp\sigma^2\log(1/\delta)}{m} + o(\frac{1}{m}). \tag{11}$$

In contrast, consider any passive learner that receives $m$ labeled examples and outputs a predictor $\hat{w}$. Consider the estimator for $\eta$ defined by $\hat{\eta} = \frac{\hat{w}}{p\alpha^2}$. $\hat{\eta}$ estimates the mean of a Gaussian distribution with variance $\sigma^2/\alpha^2$. The minimax optimal rate for such an estimator is $\frac{\sigma^2}{\alpha^2 n}$, where $n$ is the number of examples with $X = \alpha$.[3] With probability at least $1/2$, $n \leq 2mp$. Therefore, $\mathbb{E}_{D^m}[(\hat{\eta} - \eta)^2] \geq \frac{\sigma^2}{4\alpha^2 mp}$. It follows that $\mathbb{E}_{D^m}[L(\hat{w}) - L_\star] = \mathbb{E}_{D^m}[(\hat{w} - w)^2] = p^2\alpha^4 \cdot E[(\hat{\eta} - \eta)^2] \geq \frac{p\alpha^2\sigma^2}{4m} = \frac{\sigma^2}{4m}$. Comparing this to Eq. (11), one can see that the ratio between the rate of the best passive learner and the rate of the active learner approaches $O(1/p)$ for large $m$.

## 7 Discussion

Many questions remain open for active regression. For instance, it is of particular interest whether the convergence rates provided here are the best possible for this model. Second, we consider here only the plain vanilla finite-dimensional regression, however we believe that the approach can be extended to ridge regression in a general Hilbert space. Lastly, the algorithm uses static allocation of samples to stages and to partitions. In Monte-Carlo estimation Carpentier and Munos (2012), dynamic allocation has been used to provide convergence to a pseudo-risk with better constants. It is an open question whether this type of approach can be useful in the case of active regression.

## Footnotes

[1]This is a slight variation of the original result of Hsu and Sabato (2014), see Appendix A.

[2]Using the fact that $m \geq O(d \log(d) \log(1/\delta_3))$ implies $m \geq O(d \log(m) \log(1/\delta_3))$.

[3]Since $|\eta| \leq \frac{\sigma}{\alpha}$, this rate holds when $\frac{\sigma^2}{n} \ll \frac{\sigma^2}{\alpha^2}$, that is $n \gg \alpha^2$. (Casella and Strawderman, 1981)

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
