[Supplementary Material]

# Active Regression by Stratification

## Appendices

**Sivan Sabato**
Department of Computer Science
Ben Gurion University, Beer Sheva, Israel

**Remi Munos**
INRIA
Lille, France

## A  On the Derivation of Theorem 2.1

Theorem 2.1 is a useful variation of the results in Hsu and Sabato (2014). It stems from a slight change to Theorem 1 in Hsu and Sabato (2014), such that instead of requiring their 'Condition 1', which leads to the requirement: $n >= d\log(1/\delta)$, we require a bounded condition number $R$, which leads to the requirement: $n >= cR^2\log(c'R)\log(1/\delta)$, similarly to the proof of Theorem 2 there. We use the slightly stronger condition $n >= cR^2\log(c'n)\log(c''/\delta)$, with $n$ on both sides (and different constants $c, c', c''$), since it is more convenient in the derivations that follow. Note that both conditions are equivalent up to constants.

## B  Sampling according to $P_\phi$

Sampling $m$ labeled examples according to $P_\phi$ can be done by actively querying $m$, labels via standard rejection sampling. The algorithm is brought here for completeness.

---
**Algorithm 2** Sampling according to $P_\phi$

---
**input**  Sample size $m$, $\phi : \mathrm{supp}_X(D) \to \mathbb{R}_+^*$ such that $\mathbb{E}[\phi(\mathbf{x})] = 1$.
**output**  A labeled sample $S$ of size $m$ drawn according to $P_\phi$.
 1: **while** $|S| < m$ **do**
 2:     Draw $\mathbf{x}$ according to $D_X$
 3:     Draw a uniform random variable $u \sim U[0,1]$
 4:     **if** $u \le \phi(\mathbf{x})/\max_{\mathbf{z}\in\mathrm{supp}_X(D)}\phi(\mathbf{z})$ **then**
 5:         Draw $y$ according to $D_{Y|\mathbf{x}}$
 6:         $S \leftarrow S \cup \{(\mathbf{x}/\sqrt{\phi(\mathbf{x})}, y/\sqrt{\phi(\mathbf{x})})\}$.
 7:     **end if**
 8: **end while**

---

## C  Proof of Lemma 3.1

*Proof of Lemma 3.1.*  Denote $\xi := \frac{c\log(c'n)\log(1/\delta)}{n}$. Let $\beta \ge 0$, and $H_\beta = \{\mathbf{x} \mid \psi(\mathbf{x}) \le \beta\|\mathbf{x}\|_*^2\}$. There exists a $\beta \ge 0$ such that the solution for Eq. (4) has the following form.

$$\phi^\star(\mathbf{x}) = \max\{\|\mathbf{x}\|_*^2\xi, \frac{\psi(\mathbf{x})(1 - \mathbb{E}[\|\mathbf{X}\|_*^2\xi \cdot \mathbb{I}[\mathbf{X} \in H_\beta]])}{\mathbb{E}[\psi(\mathbf{X}) \cdot \mathbb{I}[\mathbf{X} \notin H_\beta]]}\}.$$

Therefore $\phi^\star(\mathbf{x}) \ge \psi(\mathbf{x})(1 - \mathbb{E}[\|\mathbf{X}\|_*^2] \cdot \xi)/\mathbb{E}[\psi(\mathbf{X})]$. Plugging this into the definition of $\rho$, and using Eq. (1),

$$\rho(\phi^\star) = \mathbb{E}[\psi^2(\mathbf{x})/\phi^\star(\mathbf{x})] \le \frac{\mathbb{E}^2[\psi(\mathbf{x})]}{1 - d\xi} \le \mathbb{E}^2[\psi(\mathbf{x})] + \frac{d\xi}{1 - d\xi} \cdot \mathbb{E}^2[\psi(\mathbf{x})].$$

For $n \ge O(d\log(d)\log(1/\delta))$, $d\xi \le 1/2$, hence $\frac{d\xi}{1-d\xi} \le 2d\xi \le O(d\log(n)\log(1/\delta)/n)$. Therefore $\rho(\phi^\star) \le \mathbb{E}^2[\psi(\mathbf{x})](1 + O(d\log(n)\log(1/\delta)/n))$. To see that $\rho(\phi^\star) \ge \mathbb{E}^2[\psi(\mathbf{x})]$, consider Eq. (4) for $\xi = 0$. In this case the optimal solution is $\phi^\star(\mathbf{x}) = \psi(\mathbf{x})/\mathbb{E}[\psi(\mathbf{x})]$.  $\square$

# D    Proof of Lemma 5.4

*Proof of Lemma 5.4.* By the definition of $\mu_i$ and $Q_i$,

$$\begin{aligned}
\mu_i &= \int_{A_i \times \mathbb{R}} \|\mathbf{X}\|_*^2 (\mathbf{X}^\top \mathbf{w}_\star - Y)^2 \, dD(\mathbf{X}, Y) \\
&= \int_{A_i \times \mathbb{R}} (\frac{\mathbf{X}^\top}{\|\mathbf{X}\|_*} \mathbf{w}_\star - \frac{Y}{\|\mathbf{X}\|_*})^2 \|\mathbf{X}\|_*^4 \cdot dD(\mathbf{X}, Y) \\
&= \Theta_i \cdot \int (\mathbf{X}^\top \mathbf{w}_\star - Y)^2 \cdot dQ_i(\mathbf{X}, Y) \\
&= \Theta_i \cdot \mathbb{E}_{Q_i}[(\mathbf{X}^\top \mathbf{w}_\star - Y)^2].
\end{aligned} \tag{12}$$

Assume that $\mathcal{E}$ holds. By Eq. (10), for all $\mathbf{X} \in \operatorname{supp}_X(Q_i)$,

$$(\mathbf{X}^\top \mathbf{w}_\star - Y)^2 \le (|\mathbf{X}^\top \mathbf{w}_\star - \mathbf{X}^\top \hat{\mathbf{v}}| + |\mathbf{X}^\top \hat{\mathbf{v}} - Y|)^2 \le (|\mathbf{X}^\top \hat{\mathbf{v}} - Y| + \Delta)^2.$$

From Eq. (12) and the definition of $\nu_i$, it follows that $\mu_i \le \nu_i$. For the upper bound on $\nu_i$,

$$\begin{aligned}
(|\mathbf{X}^\top \hat{\mathbf{v}} - Y| + \Delta)^2 &\le (|\mathbf{X}^\top \mathbf{w}_\star - Y| + |\mathbf{X}^\top \mathbf{w}_\star - \mathbf{X}^\top \hat{\mathbf{v}}| + \Delta)^2 \\
&\le (|\mathbf{X}^\top \mathbf{w}_\star - Y| + 2\Delta)^2
\end{aligned} \tag{13}$$

By Jensen's inequality, $\mathbb{E}_{Q_i}[(|\mathbf{X}^\top \mathbf{w}_\star - Y| + 2\Delta)^2] \le (\sqrt{\mathbb{E}_{Q_i}[(\mathbf{X}^\top \mathbf{w}_\star - Y)^2]} + 2\Delta)^2$. Therefore

$$\begin{aligned}
\nu_i &\equiv \Theta_i \cdot \mathbb{E}_{Q_i}[(|\mathbf{X}^\top \hat{\mathbf{w}} - Y| + \Delta)^2] \\
&\le \Theta_i (\sqrt{\mathbb{E}_{Q_i}[(\mathbf{X}^\top \mathbf{w}_\star - Y)^2]} + 2\Delta)^2 \\
&= (\sqrt{\mu_i} + 2\Delta\sqrt{\Theta_i})^2.
\end{aligned}$$

$\square$