[Reviews · NeurIPS 2014]

Submitted by Assigned_Reviewer_29

This work studies the problem of active learning in linear regression.
Unlike classification, the objective in active regression is merely
to improve the constant factors in the distribution-dependent rates
of convergence compared to passive learning, since it is known that
the asymptotic dependence on the number of labels typically cannot be
improved compared to passive learning, and nor can the worst-case
values of the constant factor. The paper argues that there is a
distribution-dependent constant factor in the rate of convergence
of passive learning, which can sometimes be improved for active
learning.

Specifically, they propose a rejection sampling scheme, which alters
the sampling distribution to a more-favorable one, without changing
the optimal solution. However, the rejection sampler requires a sort
of scaling function \phi as a parameter, and obtaining good performance
guarantees requires one to set this function carefully, and with some
dependence on the joint distribution of (X,Y). Since this latter distribution
is unknown, the algorithm attempts to optimize the choice of \phi among
piecewise constant functions, using an estimated linear function from an
initial sample. They prove a risk bound for this method, which can be
made to approach the ``oracle'' rate (where the optimal \phi is given),
and which is provably sometimes superior to the capabilities of passive
learning methods.

Overall, this seems to be a solid contribution, which I suspect will have
a wide audience.

My one main reservation is that I would have liked to see more discussion
of the dependence on K vs \rho*_A in Theorem 5.1. There are some terms
that are increasing in K, while we would like \rho*_A to decrease toward
\rho*, which presumably requires K to grow. Thus, the trade-off between
these two quantities can affect the rates. Some examples to illustrate
how we should expect this trade-off to behave, for some reasonable
distributions and sensible partition, would be helpful.

minor comments:

One citation that is missing here is Efromovich (2005): Sequential Design
and Estimation in Heteroscedastic Nonparametric Regression.
That work studies active regression as well (though for a nonparametric class),
and also finds improvements in constant factors based on the degree of
heteroscedasticity.

I also want to echo a remark of one of the other reviewers, that the notation for
P_\phi seems a little strange to me. I believe the intention is that, denoting
by Q_\phi the measure having density \phi with respect to D, we define
P_\phi as the distribution of (X/\sqrt{\phi(X)},Y/\sqrt{\phi(X)}), for
(X,Y) \sim Q_\phi. This P_\phi should then be well-defined, and
L(w,D)=L(w,P_\phi) would then follow via the law of the unconscious statistician.
Was this the intended meaning of P_\phi?
Summary: A solid paper on improving the distribution-dependent constant factors in linear regression via active learning.

Submitted by Assigned_Reviewer_44

This paper proposes a new method for actively learning a linear regressor, under
the usual least-squares loss function. The method works roughly as follows:

1. Let's say the data lie in R^d. First, a partition of the space is chosen.
Then, each data point and response value are reweighted in a simple manner that
is uniform within each cell of the partition and is specified by a single real
number within each cell. The authors show that this reweighting leaves the
squared loss of any linear function unchanged.

2. However, different weightings yield different rates of convergence for the
linear regressor. Here the authors are very insightful in the way they use
a recent generalization bound for linear regression.

3. Finding the optimal weighting, the one that leads to the best rate of convergence,
requires labels. The authors give a parsimonious, active way to do this, and along
the way, to estimate the linear regressor.

4. Steps (1)-(3) are all for a particular partition of space. The authors suggest
picking successively finer partitions as the number of points grows.

Label complexity bounds are given for steps (1)-(3).

Comments:

This paper has a lot of novel ideas and insights: of particular interest are the
reweighting method for the distribution and the way in which the new generalization
bound for regression is exploited. The paper also presents an algorithm that can
be made reasonably practical. All in all, this is a significant advance in the state
of the art in active linear regression.

There are a few things that one could quibble about:

1. The analysis requires that the "label noise" for data point x be bounded by
O(||x||^2). It would be nice to do away with this. Still, previous work on
active regression has made far stronger assumptions on the noise.

2. No method is given for refining the partition, and there is no analysis of
the asymptotic rate that would be achieved. This doesn't bother me: in practice,
a reasonable partition could be obtained by hierarchical clustering, for instance.
Summary: A novel and insightful paper that advances the state of the art in active learning for
least-squares linear regression.

Submitted by Assigned_Reviewer_47

This paper is hard to follow and not clear. It is based on recent results by Hsu et al on loss minimization (although their theorem is stated incorrectly). The paper lacks empirical validation; how we can benefit from active learning in parametric linear regression. The derivation of main theorems are not clear to me (if not wrong).

In equation 1, ||X||_* has not been defined earlier.

Please double check theorem 2.1. For sample complexity "n" should only appear once. I believe n > c log(n) is not what Hsu et al meant.

The derivation of L(W,P) = L(W,D) (on page 3) does not seem correct to me. P is a measure and as a result the second line seems bizarre. I think the result is correct though.

It is not clear what happens to the main theorem of the paper once \Lambda_D is unbounded. This happens once the distribution is heavy-tail. I think in such situations the gap between Lemma 3.1 and Theorem 5.1 will be huge.

It is very important to support the claims through a set of experiments.

Summary: I think this work needs improvement in terms of writing and stating the results. The authors should also make a better job in terms of writing the proofs. They are not clear. A set of real-world experiments are also missing.
Author Feedback
Author rebuttal: Dear Reviewers,
Thank you for your valuable and supportive feedback. Please see below our response to specific comments.

Reviewer 1 (_29):

Q: I would have liked to see more discussion
of the dependence on K vs \rho*_A in Theorem 5.1.
A: Thank you for the suggestion, we will add specific examples in the extended version of this work.

Q: One citation that is missing here is Efromovich (2005)
A: Thank you for the pointer, we will add this reference to our literature survey.

Reviewer 2 (_44):

Q: The analysis requires that the "label noise" for data point x be bounded by
O(||x||^2). It would be nice to do away with this. Still, previous work on
active regression has made far stronger assumptions on the noise.
A: Indeed in future work we hope to reduce this dependence if possible.

Q. No method is given for refining the partition, and there is no analysis of
the asymptotic rate that would be achieved. This doesn't bother me: in practice,
a reasonable partition could be obtained by hierarchical clustering, for instance.
A: One approach for refining the partition is using a progressively refined grid over the d-dimensional space. This approach has been shown in Carpentier and Munos (2012) to provide tight asymptotic rates for Monte Carlo integration of Holder functions, and a similar technique can be used here. The explicit analysis was not included in this short version for lack of space, but will be available in the long version of this work.

Reviewer 3 (_47):

Q: In equation 1, ||X||_* has not been defined earlier.
A: This is a typo that we will fix, ||X||_* should be ||X||_{\Sigma_D^{-1}}.

Q: Please double check theorem 2.1. For sample complexity "n" should only appear once. I believe n > c log(n) is not what Hsu et al meant.
A:
As mentioned in the paper, theorem 2.1 is a useful variation of the results in Hsu and Sabato (2013). There is indeed a typo in the condition on n, it should read:
“n >= c R_D^2 log(c’n) log(c’’/delta)” instead of “n >= c R_D^2 log(c’n/delta)”. This typo affects the final result in a minor way:
the initial condition on m should be “m >= O(d log(d)log(1/delta))^{5/4}”
instead of “m >= O(d log(d/delta))^{5/4}”. We will fix the final version accordingly.

More details on the derivation: Theorem 2.1 (as corrected) stems from a slight change to theorem 19 in Hsu and Sabato (2013), such that instead of requiring their “condition 1”, which leads to the requirement:
“n >= d log(1/delta)”
we require a bounded condition number R, which leads to the requirement:
“n >= c R^2 log(c’ R) log(1/delta))”,
similarly to the proof of theorem 20 there.
In theorem 2.1 we use the slightly stronger condition “n >= cR^2 log(c’n) log(c’’/delta))”, with n on both sides (and different constants c,c’,c’’), since it is more convenient in the derivations that follow. Note that both conditions are equivalent up to constants. We will clarify this in the final version.

Q: The derivation of L(W,P) = L(W,D) (on page 3) does not seem correct to me. P is a measure and as a result the second line seems bizarre. I think the result is correct though.
A: Thank you for the comment. We agree in retrospect that the second line is unclear.
The derivation is easy for discrete measures. However, it is quite technical for general integrable measures, requiring direct application of the definition of Lebesgue integration. We therefore chose to avoid the detailed proof in this short version. We will rewrite the derivation in more detail to make it clearer.

Q: It is not clear what happens to the main theorem of the paper once \Lambda_D is unbounded. This happens once the distribution is heavy-tail. I think in such situations the gap between Lemma 3.1 and Theorem 5.1 will be huge.
A: In this work we assume that the condition number of the marginal distribution on X is bounded by R_D^2, thus X is not heavy tailed and \Lambda_D <= R_D^4 always, but it can also be much smaller. The smaller \Lambda_D is, the faster the convergence.